# Interleukin-23 Mediates Osteoclastogenesis in Collagen-Induced Arthritis by Modulating MicroRNA-223

**DOI:** 10.3390/ijms23179718

**Published:** 2022-08-26

**Authors:** Shih-Yao Chen, Ting-Chien Tsai, Yuan-Tsung Li, Yun-Chiao Ding, Chung-Teng Wang, Jeng-Long Hsieh, Chao-Liang Wu, Po-Ting Wu, Ai-Li Shiau

**Affiliations:** 1Department of Nursing, College of Nursing, Chung Hwa University of Medical Technology, Tainan 71703, Taiwan; 2Department of Orthopaedics, Ditmanson Medical Foundation Chia-Yi Christian Hospital, Chiayi 60002, Taiwan; 3Department of Biochemistry and Molecular Biology, College of Medicine, National Cheng Kung University, Tainan 70101, Taiwan; 4Department of Microbiology and Immunology, College of Medicine, National Cheng Kung University, Tainan 70101, Taiwan; 5Ditmanson Medical Foundation Chia-Yi Christian Hospital, Chiayi 60002, Taiwan; 6Department of Orthopaedics, National Cheng Kung University Hospital, College of Medicine, National Cheng Kung University, Tainan 70101, Taiwan; 7Department of Orthopaedics, College of Medicine, National Cheng Kung University, Tainan 70101, Taiwan; 8Department of Biomedical Engineering, National Cheng Kung University, Tainan 70101, Taiwan; 9Medical Device Innovation Center, National Cheng Kung University, Tainan 70403, Taiwan

**Keywords:** interleukin-23, rheumatoid arthritis, microRNA-223, collagen-induced arthritis, osteoclastogenesis

## Abstract

Interleukin-23 (IL-23) plays a pivotal role in rheumatoid arthritis (RA). IL-23 and microRNA-223 (miR-223) are both up-regulated and mediate osteoclastogenesis in mice with collagen-induced arthritis (CIA). The aim of this study was to examine the association between IL-23 and miR-223 in contributing to osteoclastogenesis and arthritis. Levels of IL-23p19 in joints of mice with CIA were determined. Lentiviral vectors expressing short hairpin RNA (shRNA) targeting IL-23p19 and lisofylline (LSF) were injected intraperitoneally into arthritic mice. Bone marrow-derived macrophages (BMMs) were treated with signal transducers and activators of transcription 4 (STAT4) specific shRNA and miR-223 sponge carried by lentiviral vectors in response to IL-23 stimulation. Treatment responses were determined by evaluating arthritis scores and histopathology in vivo, and detecting osteoclast differentiation and miR-223 levels in vitro. The binding of STAT4 to the promoter region of primary miR-223 (pri-miR-223) was determined in the Raw264.7 cell line. IL-23p19 expression was increased in the synovium of mice with CIA. Silencing IL-23p19 and inhibiting STAT4 activity ameliorates arthritis by reducing miR-223 expression. BMMs from mice in which STAT4 and miR-223 were silenced showed decreased osteoclast differentiation in response to IL-23 stimulation. IL-23 treatment increased the expression of miR-223 and enhanced the binding of STAT4 to the promoter of pri-miR-223. This study is the first to demonstrate that IL-23 promotes osteoclastogenesis by transcriptional regulation of miR-223 in murine macrophages and mice with CIA. Furthermore, our data indicate that LSF, a selective inhibitor of STAT4, should be an ideal therapeutic agent for treating RA through down-regulating miR-223-associated osteoclastogenesis.

## 1. Introduction

MicroRNAs (miRNAs) have widely emerged as critical factors for the pathogenesis of rheumatoid arthritis (RA). One among them, miR-223, a bona fide molecule in mediating osteoclastogenesis, is overexpressed in fibroblast-like synoviocytes (FLSs), synovial fluid, and naive CD4^+^ T lymphocytes from patients with RA as compared to those with osteoarthritis (OA) and healthy individuals [1]. It has been demonstrated in our previous study that silencing of miR-223 prevents osteoclastogenesis and collagen-induced arthritis (CIA) in mice with concomitant increases in nuclear factor 1A (NF1A) levels and decreases in macrophage colony-stimulating factor receptor (M-CSF) levels [2]. However, much remains to be learned about the regulation of miR-223 in autoimmune arthritis. 

Interleukin (IL)-23, a heterodimeric protein composed of p40 and p19 subunits, mediates chronic joint inflammation by maintaining the survival of T helper 17 (Th17) cells in concert with induction of tumor necrosis factor (TNF), IL-1β, and IL-6 in RA FLSs [3]. Expression of the p19 subunit was increased in the synovial tissue, fluid, and peripheral blood mononuclear cells of patients with RA compared with those in patients with OA [3]. It is well-documented that the p19 subunit is positively correlated with the pathogenesis of RA by the evidence showing resistance to CIA in IL-23p19-knockout mice [4]. IL-23 can also mediate osteoclastogenesis by up-regulation of receptor activator of NF-κB (RANK) expression in myeloid precursor cells [5]. Given these critical points in RA, a phase II clinical trial of an IL-23p19 neutralization has been completed [6].

According to their roles in promoting osteoclastogenesis and arthritis, this study was undertaken to determine the interplay between IL-23 and miR-223. We first analyzed the promoter region of the mouse primary miR-223 (pri-miR-223) and found that it contained the binding sites of signal transducers and activators of transcription 4 (STAT4), which was a well-known intracellular signaling transducer of the IL-23 signaling [7,8]. Subsequent investigations were employed to prove that osteoclastogenesis and miR-223 expression could be blocked in IL-23-treated bone marrow-derived macrophages (BMMs) under the presence of miR-223 sponge and short-hairpin RNA (shRNA) specific to STAT4. Importantly, our in vivo findings revealed that amelioration of arthritis in mice with CIA by intraperitoneal injection of lentivirus-mediated shRNA gene targeting IL-23p19 (LVshIL-23p19) occurred concomitantly with down-regulation of miR-223. Furthermore, lisofylline (LSF), which is mainly a selective inhibitor of STAT4 and used to treat autoimmune diabetes [9], dampens arthritis in our mouse model by reducing miR-223-associated osteoclastogenesis.

## 2. Results

### 2.1. Amelioration of Arthritis by IL-23p19 Silencing Concomitantly with Down-Regulation of miR-223 in Mice with CIA

To assess that IL-23p19, which has been recognized as a crucial factor in the pathogenesis of RA, also plays a pivotal role in our animal model, a time course study was performed. The study observed that expression of IL-23p19 was induced on day 10 and elevated over time in the joints of mice during the progression of CIA, as determined by reverse transcription polymerase chain reaction (RT-PCR) (Figure 1A). We further silenced IL-23p19 in mice with CIA by administering intraperitoneal injections of LVshIL-23p19 and observed lower arthritis scores than those in mice receiving LVshLuc, serving as the control vector, and medium-treated control mice (Figure 1B). Histopathologic analysis of joints in LVshIL-23p19-injected mice revealed milder synovial hyperplasia, and bone and cartilage erosion, when compared to control mice (Figure 1C). To determine whether amelioration of arthritis in mice with CIA in which IL-23 was silenced was associated with interference of IL-23 downstream signaling and miR-223 levels, the expression of phosphorylated signal transducers and activators of transcription 4 (p-STAT4) and miR-223 were analyzed by immunohistochemistry and in situ hybridization. Prospectively, reduced expression of p-STAT4 concomitantly with decreased levels of miR-223 was observed in the joints of arthritic mice treated with LVshIL-23p19 (Figure 1D,E), suggesting that IL-23 signaling positively regulates miR-223 in mice with CIA.

### 2.2. IL-23 Signaling Promotes Osteoclastogenesis in BMMs through miR-223

According to our previous study, miR-223 was shown to play a key role in promoting osteoclastogenesis during the progression of CIA and might serve as a therapeutic target in RA [2]. At present, our results implied that IL-23 was an upstream regulator of miR-223 and responsible for the pathogenesis of CIA. Therefore, we anticipated that the IL-23 signaling might induce osteoclastogenesis through miR-223. Indeed, IL-23 dose-dependently increased the number of tartrate-resistant acid phosphatase (TRAP)-positive cells in BMMs from DBA/1 mice (Figure 2A). To correlate IL-23 signaling with miR-223 in osteoclastogenesis, we silenced miR-223 and STAT4 in IL-23-treated BMMs by lentiviral gene delivery of miR-223 target sequences (LVmiR-223T) [2] and shRNA specific to STAT4 (LVshSTAT4#1 and LVshSTAT4#2) and observed lower numbers of TRAP-positive cells than those in control cells (Figure 2B,C).

### 2.3. Up-Regulation of miR-223 by IL-23 Signaling in Murine Macrophages

Our findings at present suggested that IL-23 promoted osteoclastogenesis and exacerbated arthritis signs by increasing the levels of miR-223 in BMMs and mice with CIA. Therefore, we tried to dissect the mechanism in which miR-223 is regulated by predicting the molecules that bind to the promoter region of pri-miR-223. Interestingly, our analysis revealed three putative STAT4 binding sites between 126 and 149 bp upstream of the transcriptional start site of the mouse pri-miR-223 by predicting with the PROMO software [7,8]. To prove that IL-23 regulates the promoter activity of miR-223 via its downstream signaling, we treated Raw 264.7 cells with 20 ng/mL of IL-23 and observed the enrichment of phosphorylated STAT4 to the pri-miR-223 promoter, as determined by quantitative chromatin immunoprecipitation (qChIP) assay (Figure 2D). This showed that miR-223 was transcriptionally activated by IL-23 and we further confirmed this by treating BMMs with various concentrations of IL-23 and STAT4-specific shRNA carried by lentiviral vectors (LVshSTAT4). Indeed, IL-23 dose-dependently increased the levels of miR-223 in BMMs (Figure 2E), and this phenomenon was abrogated in LVshSTAT4-transfected cells compared with the two control cells in response to IL-23 stimulation, as determined by qRT-PCR (Figure 2F). Taken together, these findings indicate that IL-23 signaling through STAT4 transactivates miR-223 in murine macrophages.

### 2.4. LSF Treatment Ameliorates Arthritis through Down-Regulating miR-223-Associated Osteoclastogenesis in BMMs and Mice with CIA

Our results strongly implicated that STAT4 intermediated IL-23 signaling and miR-223-associated osteoclastogenesis. To further prove this concept in vitro, we treated BMMs with a STAT4 selective inhibitor LSF in response to IL-23 stimulation and observed their levels of osteoclastogenesis and miR-223 expression. Prospectively, osteoclast numbers and miR-233 levels were reduced in LSF-treated BMMs upon stimulating with IL-23 (Figure 3A,B). To associate our findings in vivo, we treated arthritic mice intraperitoneally with LSF on day 22 after collagen induction for two weeks. Upon euthanization, LSF-injected mice revealed lower arthritis scores and incidence than those in PBS-treated control mice (Figure 3C). Histopathologic analysis of joints in LSF-injected mice revealed milder synovial hyperplasia, and bone and cartilage erosion, as well as reduction of phosphorylated STAT4 levels and osteoclast numbers, as determined by immunohistochemical and TRAP staining, when compared to PBS-treated mice (Figure 3D,F,G). Furthermore, the LSF-treated mice with CIA also showed decreased levels of miR-223 in their joints (Figure 3E), indicating that inhibiting STAT4 activity ameliorates arthritis concomitantly with down-regulation of miR-223 and osteoclastogenesis.

## 3. Discussion

In the present study, we demonstrate for the first time that IL-23 signaling through STAT4 up-regulates miR-223 in murine macrophages and joints of mice with CIA. We further suggest that IL-23 promotes osteoclastogenesis by elevating the levels of miR-223, which plays a pivotal role in osteoclast differentiation and arthritis induction [2]. Our present data also reveal that silencing IL-23p19 by lentivirus-based shRNA gene delivery ameliorated arthritis in the mouse CIA model concomitantly with down-regulation of miR-223 (Figure 4).

IL-23 has been recognized as a potent inflammatory cytokine and mediates arthritis progression by promoting the differentiation and maintenance of Th17 cells in conjunction with the release of IL-17, TNF, and IL-6 from Th17 cells, neutrophils, myeloid cells, and RA FLSs [3]. IL-23-deficient mice were highly resistant to CIA and expressed lower levels of Th17 cells [4]. This suggested that IL-23 might play a role in the pathogenesis of RA dependent mostly upon the production of Th17 cells. We confirmed the concept in the animal study by dampening arthritis with lentivirus-based shRNA gene delivery to silence the expression of IL-23p19 in mice suffering from CIA. This strategy shows prominent therapeutic effects in reducing arthritis and histologic score in IL-23p19-silenced arthritic mice (Figure 1B,C). However, translation of the murine findings to humans was not straightforward, as anti-IL-23 inhibitors did not work in human RA [10]. However, they do show efficacy in other inflammatory diseases, including psoriatic arthritis (PsA) and ankylosing spondylitis (AS) [10]. This might be due to various reasons: First, the role of Th17 cells in the pathogenesis of RA remained controversial. Yamada and colleagues showed no difference in the frequency of Th17 cells between patients with established RA and healthy controls, and the frequency did not correlate with a 28-joint disease activity score (DAS28) [11]. Moreover, Th1 cells were more abundant than Th17 cells in the joints of RA patients [11]. Although the discovery of Th17 cells revolutionized the rheumatology fields and helped to explain some controversies of the Th1-Th2 theory [12,13], the discrepancy between these results could be attributable to the heterogeneity of RA patients. Second, patients with PsA, psoriasis, and AS shared the same variants of the IL-23 receptor, suggesting similar pathogenic mechanisms in the three diseases [14]. Third, activation of the IL-23/IL-17 pathway was the main manifestation of bone changes in both PsA and AS, however, the anti-citrullinated protein antibody (ACPA) levels were shown to promote bone loss in RA but not in PsA and AS [15,16]. Interestingly, we discovered the possibility that IL-23 might be associated with osteoclastogenesis in the CIA joint by modulating the levels of miR-223. Further investigation into the levels of ACPAs in the present experimental settings might solve the puzzle. 

Various findings have uncovered many deregulated miRNAs in synovial tissue and synovial fibroblasts (SFs) from RA patients, indicating such molecular abnormalities may contribute to the pathogenetic mechanism of RA. For instance, miR-16 expression was up-regulated in peripheral blood mononuclear cells (PBMCs) from RA patients, compared with healthy controls, and correlated with their disease activity and Th17/Treg imbalance [17]. MiR-146a expression is up-regulated in PBMCs and Th17 cells from RA patients compared with healthy donors and OA patients, and its expression is critical for regulating TNF production [18,19]. In peripheral blood, miR-155 expression was increased in both plasma and PBMCs from RA patients compared to healthy donors [20]. In addition, the miR-140 expression pattern has been well-documented in the rheumatoid joint in our previous studies [21], indicating decreased expression of miR-140 in both SFs and synovial tissue from RA patients and CIA mice. MiR-223 has been shown by Fulci and colleagues to be the only miRNA that is markedly up-regulated in RA peripheral T lymphocytes as compared to healthy controls, mostly in the naïve CD4^+^ T cell population [22]. Within these findings, we are the first to demonstrate that inhibition of miR-223 activity by miRNA sponge in the joints of CIA mice results in prevention of osteoclastogenesis [2]. MiR-223 is a potent osteoclastogenic microRNA [23] and plays pathogenic roles in RA [1]. It has been shown to express higher levels in RA FLSs than in OA FLSs [1]. Nevertheless, little is known about the regulation of miR-223 within the cytokine network in autoimmune arthritis. Here, we showed for the first time that IL-23 was an extracellular stimulator responsible for the up-regulation of miR-223 in murine macrophages, and the effect was through STAT4. Interestingly, a dramatic drop in luciferase activity between −223 to −77 bp upstream of the transcriptional start site of the mouse pri-miR-223 was observed, and 3 putative STAT4 binding sites were predicted in this promoter region [7,8]. Therefore, we anticipate that STAT4 may transcriptionally activate miR-223 by binding to the promoter region of pri-miR-223. This proposal requires a detailed investigation by reporter assay to provide evidence examining a direct regulation of the miR-223 promoter by IL-23.

Our results are in line with a previous finding demonstrating that IL-23 induces osteoclastogenesis in murine BMMs [5]. Nevertheless, the detailed mechanisms of this regulating process remain elusive. Therefore, it would be intriguing to decipher if IL-23 signaling through STAT4 could induce the expression of miR-223 during osteoclastogenesis. This hypothesis was proved by our experiments showing reduced osteoclastogenesis in the miR-223 sponge and STAT4-specific shRNA-treated BMMs upon stimulating with IL-23 (Figure 2B,C). In addition, the expression levels of miR-223 were increased in response to IL-23 stimulation dose dependently in BMMs (Figure 2E), and decreased after STAT4 silencing (Figure 2F). Interestingly, suppression of CIA is revealed in STAT4-deficient mice [24]. Further efforts to address the mechanisms that regulate the expression of miR-223 in STAT4-silenced arthritic animals are worthwhile. Therefore, we treated CIA mice with a STAT4-selective inhibitor LSF and observed that LSF profoundly reduced arthritis (Figure 3C), raising the possibility that STAT4 was likely a crucial target of LSF for its anti-arthritis effect.

LSF has been shown to exert its anti-inflammatory effects by inhibiting the production of TNF-α, IL-1β, IL-6, macrophage inflammatory protein-lα, TGF-β and IFN-γ, supporting its roles in immunoregulation [9]. LSF was originally known to block the biological effects of IL-12 that are involved in the development of Thl cell-mediated autoimmune diseases, such as multiple sclerosis [25] and type 1 diabetes [26]. The ability of LSF to inhibit IL-12 signaling by blocking the activation of STAT4 has been demonstrated in mouse allergic encephalomyelitis (EAE), an experimental model of multiple sclerosis [25]. The reduction of EAE by LSF treatment is associated with the inhibition of Thl cell differentiation and IFN-γ production [25]. RA is a known Thl cell-mediated disease [27] and the effect of LSF on this autoimmune disorder has not been elucidated before. In the present study, we proved and proposed a new mechanism by which LSF could ameliorate arthritis in mice with CIA accompanied by down-regulation of miR-223-associated osteoclastogenesis through possibly rescuing NF1A expression levels [2]. LSF might be a potential therapeutic agent for treating autoimmune arthritis in the future.

## 4. Materials and Methods

### 4.1. Construction of Lentiviral Vectors

IL-23p19 shRNA-expressing pLKO.1-IL-23p19 (TRCN0000067118, TRCN0000067119, TRCN0000067120, TRCN0000067121, and TRCN0000067122), STAT4 shRNA-expressing pLKO.1-STAT4 (TRCN0000081638, TRCN0000081639, TRCN0000081640, andTRCN0000081641) and luciferase shRNA-expressing pLKO.1-shLuc (TRCN0000072246) lentiviral plasmids were purchased from the National RNAi Core Facility (Academia Sinica, Taibei, Taiwan). Lentiviral construct pWPXL/miR-223T encoding miR-223 target sequences that served as a miRNA sponge was derived as previously described [2]. Recombinant lentiviral vectors, LVshIL-23p19, LVshSTAT4#1, LVshSTAT4#2, LVmiR-223T, and LVshLuc, were produced and their titers determined as described before [2]. Based on the results of RT-PCR analysis, we choose pLKO.1-shIL-23p19 (TRCN0000067118), pLKO.1-shSTAT4#1 (TRCN0000081638) and pLKO.1-shSTAT4#2 (TRCN0000081639) to generate lentiviral vectors encoding IL-23p19 and STAT4 shRNA.

### 4.2. Animal Studies

CIA was induced in male DBA/1 mice (aged 8~11 weeks) by immunization with bovine type II collagen and Freund’s complete adjuvant, as described previously [2]. All animal experiments were performed based on the guidelines approved by the Institutional Animal Care and Use Committee of National Cheng Kung University (Approval number: 103127). On day 21 after immunization with type II collagen, groups of 6 mice received intraperitoneal injections of LVshIL-23p19, LVshLuc (1 × 10^8^ relative infection units), and vehicle medium (Dulbecco’s modified Eagle’s medium). Another animal experiment was performed by treating arthritic mice with daily intraperitoneal injections of LSF (*n* = 6, 50 mg/kg of body weight) and phosphate-buffered saline (PBS, *n* = 7) on day 22 after collagen induction for 14 days. Arthritis signs were graded with arthritis scores (0–4 for each paw and feet; maximum possible score 16), as published previously [21,28].

### 4.3. RT-PCR, qRT-PCR, Immunohistochemistry, and In Situ Hybridization (ISH)

Total RNA from mice joint tissue was isolated with TRIzol reagents (Invitrogen/ThermoFisher Scientific, Waltham, MA, USA), and cDNA was synthesized by using a Reverse-iT First-strand Synthesis kit (ABgene/Thermo Fisher Scientific, Waltham, MA, USA) for RT-PCR with primer pairs specific to IL-23p19 (forward5′-GCAGATCACAGAGCCAGCCA-3′ and reverse 5′-CTCTAGCCAGCTGCCTGCTC-3′), and GAPDH (forward 5′-GCCATCACTGCCACCCAG-3′ and reverse 5′-TCTTACTCCTTGGAGGCCATGT-3′). Mature miR-223 was quantified with a TaqMan MicroRNA assay kit (Applied Biosystems/Thermo Fisher Scientific, Waltham, MA, USA), and qRT-PCR was performed using TaqMan Universal PCR Master Mix (Applied Biosystems/ThermoFisher Scientific, Waltham, MA, USA), as described previously [2]. Ankles from mice with CIA were snap-frozen and embedded in paraffin. The sections were deparaffinized in xylene, dehydrated in alcohol, treated with proteinase K, washed with PBS-buffered H_2_O_2_, and stained with antibodies against phosphorylated STAT4 (Abcam, Cambridge, UK). After deparaffinization, the sections for ISH were fixed in paraformaldehyde, treated with proteinase K, and acetylated with acetic anhydride in triethanolamine hydrochloride. After washing in PBS, sections were incubated in prehybridization buffer (BioChain, Newark, CA, USA) and hybridized with LNA digoxigenin (DIG)-labeled probes (EXIQON, Woburn, MA, USA) for miR-223 and scramble as a negative control. Alkaline phosphatase-conjugated anti-DIG antibody was then incubated and stained with 5-Bromo-4-chloro-3-indolyl phosphate (BCIP)/nitroblue tetrazolium (NBT). The signal intensity was further quantitated using Image J version 1.42q (National Institutes of Health), as described previously [10].

### 4.4. Quantitative Chromatin Immunoprecipitation (qChIP) and TRAP Staining

QChIP was performed with the EZ-ChIP kit (Merck Millipore, Boston, MA, USA) according to the manufacturer’s instructions. Briefly, IL-23-treated RAW264.7 cells were formaldehyde crossed-linked, and cell lysates were sonicated to shear the DNA to an average length of 500 bp followed by immunoprecipitation with rabbit anti-phosphorylated STAT4 (ChIP grade, Cell Signaling Technology, Danvers, MA, USA) antibody and normal rabbit IgG (Santa Cruz Biotechnology, Dallas, TX, USA) in combination with protein G agarose beads. The immunoprecipitated protein-DNA crosslinks were reversed by heating. The following primers were used for qRT-PCR of immunopurified DNA as well as input samples: 5′-GGTTGAGAATGGGTGGAGAC-3′ (forward) and 5′-TGTGAGCAGGAAGGTCACAT-3′ (reverse) corresponding to the mouse pri-miR-223 promoter as published before [8]. Values were normalized to the input by using: % Input = 2^Ct(input)−Ct(samples)^. TRAP staining was performed using a leukocyte acid phosphatase kit (Sigma-Aldrich, St. Louis, MO, USA). The number of TRAP-positive multinucleated osteoclasts from BMMs after RANKL stimulation and various treatments were counted as published previously [2].

### 4.5. Statistical Analysis

Differences in arthritis score and incidence were compared by repeated-measures analysis of variance and chi-squared test. Other data were compared by Student’s *t*-test. *p* values less than 0.05 were considered significant.

## 5. Conclusions

In conclusion, with the use of murine macrophages and an experimental model of arthritis, we herein have demonstrated that IL-23 signaling promotes osteoclastogenesis by modulating miR-223 in the mouse arthritic joint. Although translation of murine anti-IL-23 therapeutics to humans is not straightforward, our findings provide a new mechanism that the IL-23 inhibition may inhibit bone destruction in CIA by preventing osteoclastogenesis through down-regulating miR-223.

## Figures and Tables

**Figure 1 ijms-23-09718-f001:**
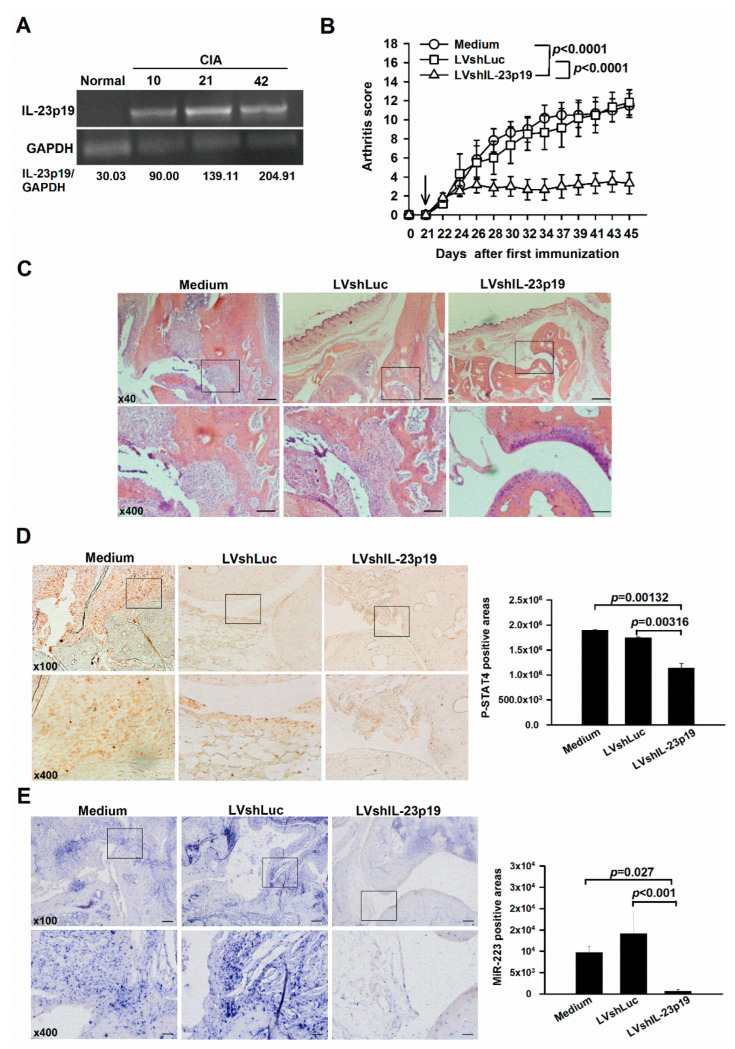
Endogenous expression of IL-23p19 and clinical assessments in the LVshIL-23p19-treated mice with collagen-induced arthritis (CIA). (**A**) Reverse transcription polymerase chain reaction (RT-PCR) analysis of IL-23p19 expression in the synovial tissue of mice with CIA (Day 10, 21, 42). Each lane represents pooled samples from 3 animals. Results are representative of two independent experiments. (**B**) Amelioration in mice with CIA by evaluating with arthritis score after the LVshIL-23p19 treatment. Mice that had been immunized with collagen on days 0 and 21 were injected intraperitoneally with LVshIL-23p19, LVshLuc and medium on day 21, respectively. The arrow indicates the time at which the lentiviral vectors were injected. Each value shown represents the mean ± SEM (*n* = 6). (**C**) Representative images of the ankle sections by hematoxylin and eosin staining on day 45. (**D**) Representative images and quantification of the levels of phosphorylated signal transducers and activators of transcription 4 (p-STAT4) by immunohistochemical staining on day 45. Each value shown represents the mean ± SEM (*n* = 3). (**E**) Representative images of the levels of miR-223 by in situ hybridization on day 45. Scale bars represent 500, 200 and 50 μm in ×40, ×100 and ×400 magnifications, respectively. Each value shown represents the mean ± SEM (*n* = 3). Black boxed areas were shown at higher magnification in the panels (×400) beneath them. Results are representative of two independent experiments.

**Figure 2 ijms-23-09718-f002:**
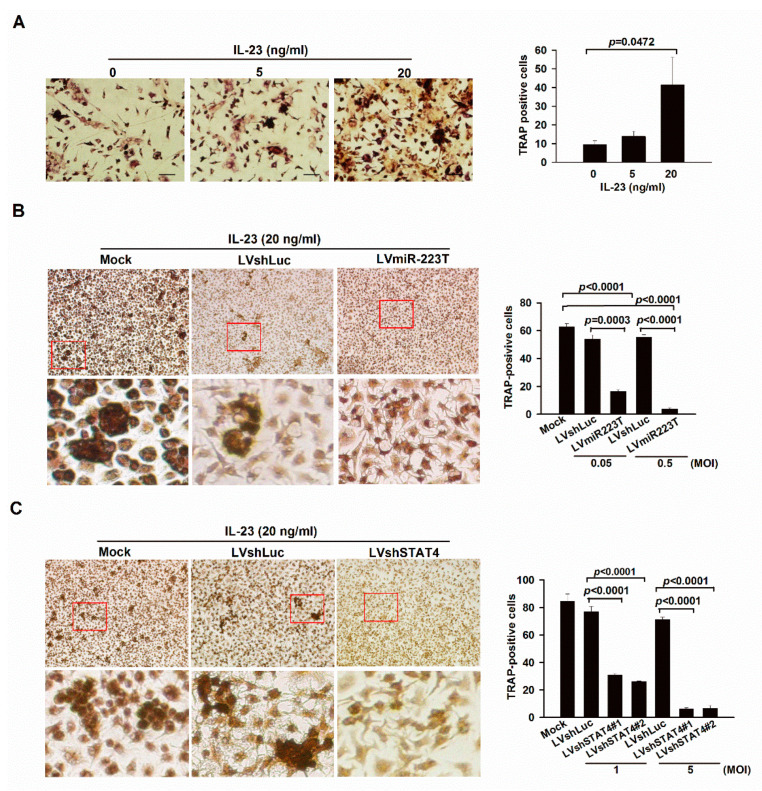
Association of IL-23 signaling with miR-223 in osteoclastogenesis. (**A**) Tartrate-resistant acid phosphatase (TRAP) staining and quantification in IL-23 (5 and 20 ng/mL) treated bone marrow-derived macrophages (BMMs). Each value shown represents the mean ± SEM (*n* = 4). (**B**) TRAP staining and quantification in LVmiR-223T and LVshLuc-infected BMMs (MOI = 0.05 and 0.5) in response to IL-23 stimulation (20 ng/mL). Each value shown represents the mean ± SEM (*n* = 3). (**C**) TRAP staining and quantification in LVshSTAT4#1 (TRCN0000081638), LVshSTAT4#2 (TRCN0000081639) and LVshLuc-infected BMMs (MOI = 1 and 5) in response to IL-23 stimulation (20 ng/mL). Scale bars represent 200 μm in ×100 magnifications. Each value shown represents the mean ± SEM (*n* = 3). (**D**) Binding of phosphorylated STAT4 to the miR-223 promoter in IL-23-treated RAW264.7 cells by determining with quantitative chromatin immunoprecipitation (qChIP) assay. Each value shown represents the mean ± SEM (*n* = 3). (**E**) Expression of miR-223 in BMMs treated with various concentrations of IL-23 for 12 h, as determined by quantitative RT-PCR (qRT-PCR). Each value shown represents the mean ± SEM (*n* = 3). (**F**) Expression of miR-223 in LVshLuc, LVshSTAT4#1 (TRCN0000081638) and LVshSTAT4#2 (TRCN0000081639)-transduced BMMs upon stimulating with IL-23 (20 ng/mL) for 12 h, as determined by qRT-PCR. Each value shown represents the mean ± SEM (*n* = 3). Red boxed areas are shown at higher magnification in the panels beneath them. Results are representative of two independent experiments.

**Figure 3 ijms-23-09718-f003:**
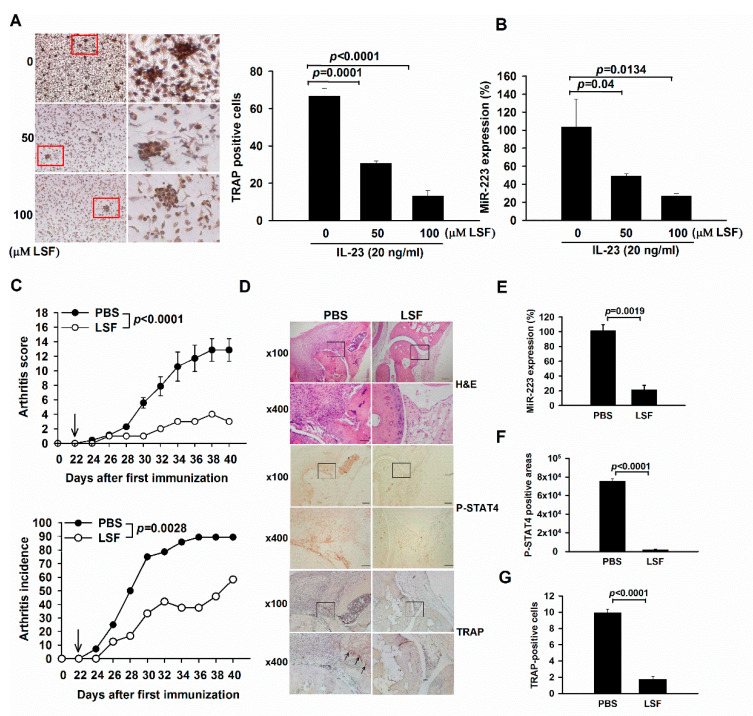
Lisofylline (LSF) treatment in BMMs and mice with CIA. (**A**) TRAP staining and quantification in LSF (50 and 100 μM)-treated BMMs upon stimulating with IL-23 (20 ng/mL). Each value shown represents the mean ± SEM (*n* = 3). Red boxed areas are shown at higher magnification in the panels next to them. Scale bars represent 200 μm in ×100 magnifications. (**B**) Expression of miR-223 in BMMs treated with various concentrations of LSF in response to IL-23 (20 ng/mL) stimulation for 12 h, as determined by qRT-PCR. Each value shown represents the mean ± SEM (*n* = 3). (**C**) Arthritis score and incidence in mice with CIA through intraperitoneal injections of LSF (50 mg/kg) and PBS daily from day 22 to 36. Arrows indicate the time at which LSF was injected. Each value shown represents the mean ± SEM (PBS: *n* = 6, LSF: *n* = 6). (**D**) Representative images of the ankle sections by hematoxylin and eosin staining on day 40. Scale bars represent 200 and 50 μm in ×100 and ×400 magnifications, respectively. Each value shown represents the mean ± SEM (*n* = 6). Black boxed areas were shown at higher magnification in the panels (×400) beneath them. Expression of (**E**) miR-223, (**F**) phospho-STAT4 (p-STAT4) and (**G**) the number of TRAP-positive cells in joint extracts of LSF and PBS-injected mice with CIA, as determined by qRT-PCR. Each value shown represents the mean ± SEM (*n* = 6). Results are representative of two independent experiments.

**Figure 4 ijms-23-09718-f004:**
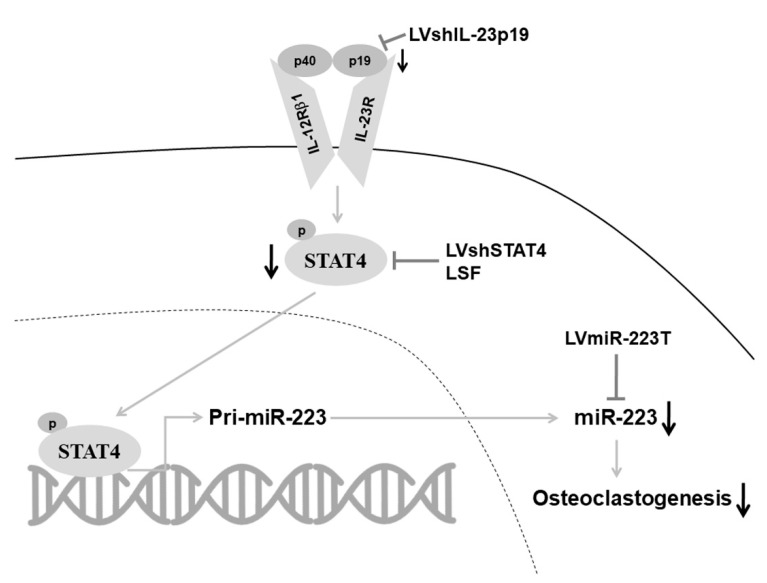
A schematic overview of silencing IL-23-STAT4-miR-223 signaling axis in preventing osteoclastogenesis. IL-23p19 can bind to the IL-23 receptor (IL-23R), which causes phosphorylation and translocation of STAT4 into the nucleus. P-STAT4 further conjugates with the promoter region of primary miR-223 (pri-miR-223) and the transactivated pri-miR-223 translocates into the cytosol to become mature miR-223 (miR-223) and mediates osteoclastogenesis. The signaling axis, IL-23-STAT4-miR-223 is proved to mediate osteoclastogenesis by introducing IL-23-stimulated BMMs and mouse CIA model with LVshIL-23p19, LVshSTAT4, LSF, and LVmiR-223T that target IL-23p19, STAT4, and miR-223.

## Data Availability

The data that support the findings of this study are available on request from the corresponding author.

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
