# Peer review of "Interleukin-23 Mediates Osteoclastogenesis in Collagen-Induced Arthritis by Modulating MicroRNA-223"

_ijms, 2022, doi:10.3390/ijms23179718_

Round 1

Reviewer 1 Report

Review for “Interleukin-23 Mediates Osteoclastogenesis in Collagen-Induced Arthritis by Modulating MicroRNA-223”

The authors have nicely shown the role of IL-23 and miR-223 in Osteoarthritis. Basing on their previous study the authors have further dissected the mechanism between IL-23 and miR-223. By using both in vitro and in vivo approaches the authors showed the role of IL-23 in promoting miR-223 expression via enhanced binding of transcription factor STAT4 to the promoter of pri-miR-223. Further by blocking STAT4 using a selective inhibitor the expression of miR-223 could be targeted and ameliorate arthritis. Overall, a good study but with few minor points to address:

1.     It would be good to include a schematic overview of all the results, for a comprehensive overview of the data obtained.

2.     Line 212: please rephrase the word “frustrating”

3.     Scale bars are missing in some of the image panels, please check and add.

4.     As miRNAs can target multiple target mRNAs simultaneously, it would be advantageous towards the development of therapy to RA, if the authors could show or discuss about the downstream targets of miR-223 that are possibly involved in preventing osteoclastogenesis.

5.     Is the expression of NF1A rescued after LSF treatment? Are these rescue effects mediated by miR-223 also involve NF1A?

6.     Fig 1a: please mention days in figure legend.

7.     Fig 1d and 1e: It would be good to replace with higher quality images. Please also describe in methods how the positive areas were estimated.

8.     Fig 2b: Images are not of good quality and are not representative of the quantification. A) Is there a reason why mock treated cells look differently morphologically in the lower panel than in the other two images. B) LVshLuc cells show fewer positive cells and lower intensity than the LVmiR223T, which is in contrast with the quantification? C) Replace with higher quality images.

9.     Fig 2c: Images should be improved.

Author Response

Comments and Suggestions for Authors

Review for “Interleukin-23 Mediates Osteoclastogenesis in Collagen-Induced Arthritis by Modulating MicroRNA-223”

The authors have nicely shown the role of IL-23 and miR-223 in Osteoarthritis. Basing on their previous study the authors have further dissected the mechanism between IL-23 and miR-223. By using both in vitro and in vivo approaches the authors showed the role of IL-23 in promoting miR-223 expression via enhanced binding of transcription factor STAT4 to the promoter of pri-miR-223. Further by blocking STAT4 using a selective inhibitor the expression of miR-223 could be targeted and ameliorate arthritis. Overall, a good study but with few minor points to address:

Response: Thank you very much for reviewing our manuscript and leaving many valuable suggestions and comments, we’ll respond to your suggestions as follows,

  1. It would be good to include a schematic overview of all the results, for a comprehensive overview of the data obtained.

Response: Thank you for this valuable suggestion. A schematic overview of all the results has been included in the new figure 4.

  1. Line 212: please rephrase the word “frustrating”

Response: “frustrating” has been replaced with “not compatible” (Line 226).

  1. Scale bars are missing in some of the image panels, please check and add.

Response: We are very sorry about this issue. All the scale bars have been checked. Some scale bars are attached in light silver, therefore, zooming in on these figures is required to observe them. In Fig. 2B, 2C, and 3A, boxed areas are shown at higher magnification in the panels beneath or next to them, therefore, there will be no scale bars in these magnified images. 

  1. As miRNAs can target multiple target mRNAs simultaneously, it would be advantageous towards the development of therapy to RA, if the authors could show or discuss about the downstream targets of miR-223 that are possibly involved in preventing osteoclastogenesis.

Response: Thank you for the comments. We have described in detail the downstream targets of miR-223, and NF1A is involved in preventing osteoclastogenesis (Line 55), according to our previous publication (reference 2).    

  1. Is the expression of NF1A rescued after LSF treatment? Are these rescue effects mediated by miR-223 also involve NF1A?

Response: Thank you for these critical comments. We suggested this might a possible mechanism. Therefore, we modified the sentence “ In the present study, we proved and proposed a new mechanism that LSF could ameliorate arthritis in mice with CIA accompanied by down-regulation of miR-223-associated osteoclastogenesis through possibly rescuing NF1A expression levels.” (Line 295-298)  

  1. Fig 1a: please mention days in figure legend.

Response: Thank you for the suggestion. Days have been included in Fig 1a legend (Line 105).

  1. Fig 1d and 1e: It would be good to replace with higher quality images. Please also describe in methods how the positive areas were estimated.

Response: Thank you for these valuable suggestions. Fig 1d and 1e have been replaced with higher quality images and the method to estimate the positive areas has also been included (Line 345-346).

  1. Fig 2b: Images are not of good quality and are not representative of the quantification. A) Is there a reason why mock treated cells look differently morphologically in the lower panel than in the other two images. B) LVshLuc cells show fewer positive cells and lower intensity than the LVmiR223T, which is in contrast with the quantification? C) Replace with higher quality images.

Response: Thank you for these critical comments. A) We suggested that lentiviral vector infection might affect the morphology of LVshLuc or LVmiR-223T-treated cells. B) It seems to us that LVshLuc cells do not show fewer positive cells and lower intensity than the LVmiR223T, which is not in contrast with the quantification. We suggest the aggregated and multi-nucleated cells will be the osteoclasts. C) Due to time constraints, it will be difficult for us to repeat the data. We suggested that zooming in on the images could clearly observe the TRAP-positive cells.    

  1. Fig 2c: Images should be improved.

Response: Thank you for the suggestion. As the response to you above, time constrains us to repeat the data, we suggested that zooming in on the images could observe them clearly, especially the magnified images could help to identify the aggregated, multi-nucleated, and TRAP-positive osteoclasts. 

Reviewer 2 Report

The authors demonstrated that IL-23 signaling upregulates miR-223 involving STAT4 that affects the number of TRAP-positive cells in murine macrophages.

Comments

1.      Lines 28,32,35,42: all the abbreviations should be disclosed on first use.

2.      Lines 57-63; 84-87: These sentences are not clear. They should be clarified.

3.      Fig 3a,b: The treatment agent should be indicated.

4.      Lines 212, 214: References are required at the end of these sentences.

5.      Lines292,320,328,345 et al: The source of reagents should be indicated including Company, City, Country.

6.      Lines 211-212: As anti-IL-23 inhibitors do not work in RA, the authors should rephrase all their conclusions and study motivations. In addition, to conclude the effect of the examined agents on osteoclastogenesis the authors should have examined other corresponding biomarkers.

Author Response

Comments and Suggestions for Authors

The authors demonstrated that IL-23 signaling upregulates miR-223 involving STAT4 that affects the number of TRAP-positive cells in murine macrophages.

Comments

  1. Lines 28,32,35,42: all the abbreviations should be disclosed on first use.

Response: Thank you for the suggestion. We think that all the abbreviations have been disclosed on first use. For example, line 28: miR-223, please check line 25: microRNA-223 (miR-223); line 32: shRNA, please check line 29: short hairpin RNA (shRNA); line 35: primary miR-223 (pri-miR-223) is the first use here; line 42: LSF, please check line 30: lisofylline (LSF).  

  1. Lines 57-63; 84-87: These sentences are not clear. They should be clarified.

Response: We believe that these sentences have clearly clarified the role of IL-23p19 in human RA and mouse CIA with proper references (Line 58-64), according to this knowledge, we designed our experiment to silence IL-23p19 in mice with CIA for dissecting the downstream mechanisms (Line 85-88).   

  1. Fig 3a,b: The treatment agent should be indicated.

Response: Thank you very much for the suggestion. The treatment agents were all included in Fig. 3a,b.

  1. Lines 212, 214: References are required at the end of these sentences.

Response: Thank you for the suggestion. The sentences have been cited with proper references (Line 226, 228)

  1. Lines292,320,328,345 et al: The source of reagents should be indicated including Company, City, Country.

Response: Thank you very much for this important suggestion. All the source of reagents has been indicated including Company, City, and Country.

  1. Lines 211-212: As anti-IL-23 inhibitors do not work in RA, the authors should rephrase all their conclusions and study motivations. In addition, to conclude the effect of the examined agents on osteoclastogenesis the authors should have examined other corresponding biomarkers.

Response: Thank you for these critical suggestions. We have rephrased our conclusion as follows” In conclusion, with the use of murine macrophages and an experimental model of arthritis, we herein have demonstrated that IL-23 signaling promotes osteoclastogenesis by modulating miR-223 in the mouse arthritic joint. Although translation of murine anti-IL-23 therapeutics to humans is not straightforward, our findings provide a new mechanism that the IL-23 inhibition may inhibit bone destruction in CIA by preventing osteoclastogenesis through down-regulating miR-223.” (Line 368-373). The study motivation is based on our previous publication [2], therefore, we suggest it’s reasonable for the present study design. Having said that we have included a big paragraph discussing the reasons for failed anti-IL-23 therapy in human RA (Line 225-240). 
